# Efficient Generation of Knock-In Zebrafish Models for Inherited Disorders Using CRISPR-Cas9 Ribonucleoprotein Complexes

**DOI:** 10.3390/ijms22179429

**Published:** 2021-08-30

**Authors:** Erik de Vrieze, Suzanne E. de Bruijn, Janine Reurink, Sanne Broekman, Vince van de Riet, Marco Aben, Hannie Kremer, Erwin van Wijk

**Affiliations:** 1Department of Otorhinolaryngology, Radboud University Medical Center, 6525 GA Nijmegen, The Netherlands; sanne.broekman@radboudumc.nl (S.B.); Vince.vandeRiet@student.ru.nl (V.v.d.R.); Hannie.Kremer@radboudumc.nl (H.K.); Erwin.vanwyk@radboudumc.nl (E.v.W.); 2Donders Institute for Brain Cognition and Behaviour, 6500 GL Nijmegen, The Netherlands; Suzanne.deBruijn@radboudumc.nl (S.E.d.B.); Janine.Reurink@radboudumc.nl (J.R.); marco.aben@wur.nl (M.A.); 3Department of Human Genetics, Radboud University Medical Center, 6525 GA Nijmegen, The Netherlands

**Keywords:** zebrafish, CRISPR-Cas9, knock-in, homology-directed repair, disease models

## Abstract

CRISPR-Cas9-based genome-editing is a highly efficient and cost-effective method to generate zebrafish loss-of-function alleles. However, introducing patient-specific variants into the zebrafish genome with CRISPR-Cas9 remains challenging. Targeting options can be limited by the predetermined genetic context, and the efficiency of the homology-directed DNA repair pathway is relatively low. Here, we illustrate our efficient approach to develop knock-in zebrafish models using two previously variants associated with hereditary sensory deficits. We employ sgRNA-Cas9 ribonucleoprotein (RNP) complexes that are micro-injected into the first cell of fertilized zebrafish eggs together with an asymmetric, single-stranded DNA template containing the variant of interest. The introduction of knock-in events was confirmed by massive parallel sequencing of genomic DNA extracted from a pool of injected embryos. Simultaneous morpholino-induced blocking of a key component of the non-homologous end joining DNA repair pathway, Ku70, improved the knock-in efficiency for one of the targets. Our use of RNP complexes provides an improved knock-in efficiency as compared to previously published studies. Correct knock-in events were identified in 3–8% of alleles, and 30–45% of injected animals had the target variant in their germline. The detailed technical and procedural insights described here provide a valuable framework for the efficient development of knock-in zebrafish models.

## 1. Introduction

The use of CRISPR-Cas9-based genome editing to create loss-of-function alleles is nowadays common practice in most model organisms. While the first reports employing the CRISPR-Cas9 system to knock-in specific genetic variants by means of homology directed repair (HDR) date back to 2013 [1,2], the overall efficiency of this application remains much lower as compared to the generation of knock-out models. In zebrafish, one of the easiest animal models to manipulate with CRISPR-Cas9, precise editing using a template for HDR was first reported by Irion and colleagues in 2014 [3]. They used a circular template vector containing 875 bp homology arms flanking the variant of interest, which they co-injected with *Cas9* mRNA and in-vitro synthesized sgRNA molecules to repair the albino mutation (*alb* allele). More recent publications report on the use of short (50–136 nt), single stranded oligonucleotide templates for HDR [4,5,6]. The use of oligonucleotide templates is advantageous compared to the use of vector templates with long homology arms, as these oligonucleotide templates can be ordered commercially. While the CRISPR-Cas9 system offers great potential for the introduction of specific genetic mutations in the zebrafish genome, there are still only few studies that employ this strategy to model the functional effects of human disease-associated genetic variants in zebrafish.

As a consequence of the implementation of next generation sequencing technologies—such as whole exome sequencing and whole genome sequencing—the number of variants of unknown significance (VUS), identified in genome diagnostics laboratories around the globe, is rapidly growing. For genes with a high sequence conservation, the zebrafish can offer an attractive model to assess the functional effects of VUS. The work of Tessadori et al., who used 50-mer symmetrical oligonucleotide templates, is a good example of modeling genetic variants orthologous to those associated with human disease in zebrafish. The percentage of animals in which the variant was introduced in the germline was found to range between 4% and 21% [5]. While the identification of founder fish in 21% of injected embryos sounds promising, they reported that the chance of success largely relies on the initial efficiency of the ribonucleoproteins (RNPs) to induce DNA cleavage, and of RNPs to cleave in close proximity to the target variant. A strong inverse relationship between knock-in efficiency and the distance of the modification to the Cas9 cut site has indeed been reported [7,8]. A recent publication showed that the generation of CRISPR-Cas9-mediated knock-in zebrafish models could be significantly improved by the use of antisense asymmetric oligonucleotides with homology arms of 36 and 90 nucleotides [4]. The 36-nt arm of the oligonucleotide is thought to hybridize with the non-target strand of the target gene, which becomes available upon binding of the Cas9-gRNA RNP complex to the target strand, and thereby facilitates homology-directed repair [9].

Introducing variants associated with genetic disorders does not always leave room to comply with the parameters for an optimal design. For example, directing Cas9 to cleave in close proximity of the target variant relies on the availability of a Cas9-specific protospacer-adjacent motif (PAM site). While variants of Cas9 with different PAM requirements are rapidly emerging, they are not yet commercially available as recombinant proteins to use in Cas9-gRNA RNP complex injections in zebrafish embryos. Techniques such as base-editing and prime-editing are often seen as the solution to this problem [10,11]. However, the studies describing these techniques have prepared their own recombinant proteins for microinjection, and/or are currently restricted to the injection of in-vitro transcribed mRNA.

Another option to improve knock-in efficiency is to downregulate or block the non-homologous end joining pathway (NHEJ) to effectively increase HDR-mediated DNA repair upon nuclease activity. The NHEJ pathway is the primary DNA repair pathway in the developing zebrafish embryo, with small insertions and deletions a result [12,13]. Downregulation of the NHEJ pathway has been reported in human cells [14], via small molecule inhibitors. We previously employed morpholino (PMO) antisense oligonucleotide-mediated knockdown of Ku70, the protein that forms a complex with Ku80 to recruit NHEJ proteins to broken DNA ends [13], to introduce a human pseudoexon into the zebrafish *ush2a* gene [15]. After the successful introduction of the pseudoexon, we further optimized our CRISPR-Cas9 protocols, and investigated the effect of Ku70 knockdown to enhance knock-in efficiency in more detail.

Here, we present our efficient procedure to develop knock-in zebrafish models using Cas9-gRNA RNP complexes and antisense asymmetric template oligos. We illustrate our approach using two variants that were identified after genetic screening of patients with hereditary sensory deficits. The first is the retinitis pigmentosa (RP) and Usher syndrome-associated c.2276G>T (p.(Cys759Phe), NM_206933) variant in *USH2A.* The pathogenicity of this variant has been under debate after it was identified in 2003 in homozygous state in two individuals with normal visual function [16,17]. The second variant is a small in-frame deletion in *RIPOR2* (c.1696_1707del; p.(Gln566_Lys569del), NM_014722.3) that we recently identified as a major cause of dominantly inherited hearing loss DFNA21 [18]. We furthermore show how morpholino-based knockdown of Ku70 increased knock-in efficiency for one of the targets. With somatic knock-in events in 3.4–18.0% of sequencing reads, and germline transmission of the variants of interest in 30–45% of adult zebrafish, our approach can provide a template for a more frequent use of zebrafish to model the functional effects of genetic variants associated with human disorders. A detailed protocol is provided in the supplemental data (Appendix A).

## 2. Results

### 2.1. sgRNA Design and Efficiency

The c.2276G>T variant in *USH2A*, leading to the substitution of a phenylalanine for cysteine at position 759 of the usherin protein, is orthologous to a c.2312G>T (p.(Cys771Phe)) substitution in zebrafish *ush2a* (Appendix A). For this variant, sgRNA design was limited by the genetic context of the target site. The six top-ranked sgRNA target sites predicted for Cas9 that are close to the variant of interest, and allow for the introduction of a silent PAM-site-disrupting variant, are 19, 32, and 47 nt downstream, and 12, 29, and 36 nt upstream of position c.2312. For an efficient introduction of the target variant, the Cas9 cleavage site should be ideally at a distance <10 nt from the target site [8]. We therefore decided to use a lower-ranked sgRNA that directs Cas9 to cleave between the second and third nucleotide upstream of position c.2312, yet might display some off-target cleavage activity. To improve in-vitro sgRNA transcription we introduced a C>G change at the 5′ position of the target-specific sequence. The resulting sgRNA has three predicted off-target sites with 1–2 nt mismatches and/or 1 nt bulge. We considered this situation preferable over the use of a more specific sgRNA that would direct Cas9 to cleave further away from position c.2312. The second variant, c.1777_1788del (p.(Gln593_Arg596del)) in *ripor2*, corresponds to the DFNA21-associated c.1696_1707del variant in the human *RIPOR2* gene (Appendix A). We identified a sgRNA that directs Cas9 to cleave the DNA between nucleotides 8 and 9 of the 12 nucleotides that we aim to delete.

Prykhozhij et al. previously showed that antisense asymmetric HDR templates provide improved knock-in efficiency over sense symmetric oligos [4]. We therefore designed asymmetric antisense HDR templates to introduce our variants of interest based on their strategy. The *ush2a* C771F HDR template includes the c.2312 G>T substitution and a silent, PAM-site disrupting C>T substitution at position c.2304 (Figure 1A). The *ripor2* del12 HDR template (shown in sense orientation) includes the c.1777_1788del variant and a silent, PAM-site disrupting G>A substitution at position c.1790 (Figure 1B).

For both targets, we first investigated whether Cas9-sgRNA ribonucleoprotein (RNP) complexes were efficient in generating lesions at the target site. For both targets we injected the appropriate RNP complexes into the first cell of fertilized zebrafish embryos (zygote). At 1 day post fertilization (dpf), we collected a subset of the injected embryos for genetic analysis (*n* = 7). To rapidly detect the presence of genomic lesions in the embryos, we extracted DNA from individual RNP-injected and uninjected embryos, and amplified a small region flanking the sgRNA target site using a nucleic acid dye-supplemented PCR. We subsequently performed high-resolution melting (HRM) analyses of the amplicons to rapidly visualize genome-editing events [19]. For both *ush2a* (Figure 1C) and *ripor2* (Figure 1D), HRM analysis of uninjected embryos revealed a single peak that corresponds with the presence of wildtype amplicons. HRM analysis of *ush2a* RNP-injected embryos resulted in the typical double peak signature of CRISPR-Cas9-induced genomic lesions (Figure 1C). The nature of both mutant and wildtype amplicons was verified using Sanger sequencing (Figure 1C, right panel). The typical overlapping sequence traces following the Cas9 cleavage site indicate the presence of multiple somatic mutations within the embryo. The *ripor2* RNP-injected embryos did not display the typical double peaks on HRM analysis, but rather a single peak with a heightened left shoulder. Sanger sequence revealed that also this pattern corresponds to the presence of multiple somatic mutations (Figure 1D, right panel). The double peaks are observed relatively far upstream from the Cas9 cleavage site (which cannot be identified in the sequencing traces). However, the 5′ boundary does not match that the intended deletion indicating that different deletions are formed upon injection of the *ripor2* RNP complex. For both RNPs, we observed genome-editing events in 100% of injected embryos.

### 2.2. Efficient Generation of Knock-In Alleles Using CRISPR-Cas9 RNPs and Asymmetric Antisense HDR Templates

From the high on-target genome-editing efficiency, we concluded that both the *ush2a* and *ripor2* RNPs are likely suitable to knock-in the variants of interest. We next injected the antisense asymmetric oligonucleotide templates together with the appropriate *ush2a* or *ripor2* RNPs into the zebrafish zygotes. To minimize the influence of differences between individual microinjections and batch-to-batch variation in egg quality, all injections were done in 4–6 replicate injections, using wildtype zebrafish embryos from 3–5 different spawnings. As the introduction of the patient-specific variants does not create a variant-specific restriction site, or allows for alternative low-cost means to assess knock-in efficiency, we immediately proceeded with massive parallel sequencing to visualize potential knock-in events. Genomic DNA was extracted from pools of 20–25 injected zebrafish embryos at 1 dpf, after which the knock-in target regions were amplified by PCR using primers upstream and downstream from the HDR template, and subjected to Ion Torrent sequencing. Since we aimed to introduce a nucleotide substation and 12-nt deletion, we assumed that the amplification bias in the Ion Torrent sequencing results was negligible. The Ion Torrent platform has a slightly higher error rate than the frequently used Illumina sequencing platform [20], which we countered with a high sequencing depth for each target (>50,000 reads obtained from the genomic DNA of each pool of embryos).

For *ush2a*, we observed 85.7 ± 8.6% of mutant (non-wildtype) reads after injection of zygotes with sgRNA-Cas9 RNP complexes and HDR template (Figure 2A). This confirms the high genome-editing efficiency that we observed by individual HRM analyses of embryos injected with *ush2a* sgRNA-Cas9 RNP complexes. We also observed high genome-editing efficiency of the sgRNA-Cas9 RNPs in *ripor2* knock-in injected embryos (80.5 ± 24.7%; Figure 2A). The large standard deviation results from one injection in which only 45% of the reads contained an indication for sgRNA-Cas9 RNP activity. To place our obtained editing-efficiency into perspective, we also injected the sgRNA for *tp53* R217H previously described by Prykhozhij et al. in their study to introduce the p.R217H variant using the antisense asymmetric oligo templates for HDR in zebrafish [4]. Note that Prykhozhij et al. made use of *Cas9* mRNA, whereas we injected in vitro assembled sgRNA-Cas9 RNP complexes. We observed 56.6 ± 1.7% (*n* = 2) of mutant *tp53* reads (Figure 2A). In the study of Prykhozhij et al., the delivery of sgRNAs with *Cas9* mRNA resulted in only 28.4% of reads in which genome-editing events were identified.

We defined successful knock-in reads as all reads that contain the variant of interest, with or without PAM-site change, and do not contain any other insertions, deletions or single nucleotide changes in the target region. Correct knock-in of the target variants was found in 5.2 ± 1.4% of the reads from embryos injected with RNP complexes and template to introduce the c.2312G>T (p.(Cys771Phe)) variant in *ush2a* (henceforth termed *ush2a* C771F injected embryos), and 12.4 ± 5.8% of the reads from embryos injected with RNP complexes and template to introduce the c.1777_1788del; p.(Gln593_Arg596del) variant in *ripor2* (*ripor2* del12 injected embryos) (Figure 2B). The majority (>75%) of the correct *ush2a* C771F knock-in reads also contained the PAM-site change. For *ripor2* del12, a little under half of the correct knock-in reads contained the PAM-site change. When expressed as percentage of the total reads, 4.0 ± 1.3% (*ush2a*) and 5.7 ± 2.9% (*ripor2*) of reads displayed both the correct knock-in of the variant and the PAM-site change. There were also many other reads in which the target variant was identified, together with additional insertions, deletions or nucleotide substitutions. These events occurred approximately 2–4 times less frequently as compared to the correct knock-in events, and are collectively displayed as knock-in events with indels (Figure 2B). Examples of the different types of reads are provided in Figure 2C,D for respectively *ush2a* C771F and *ripor2* del12. The reads including the variant of interest together with an insertion, deletion, or nucleotide substitution, are examples of erroneous genome-editing outcomes that were observed repeatedly within the sequencing reads (>50,000 reads per target). We also performed this analysis for the *tp53* R217H injected embryos (two replicate injections), and identified correct knock-in events in 4.3 ± 0.8% of reads, and correct knock-in events including the PAM-site change in 3.0 ± 0.6% of reads (Figure 2B). This percentage is slightly lower than that of *ush2a* C771F injected embryos, and markedly lower than that of *ripor2* del12 injected embryos. However, the original publication on the knock in of the *tp53* R217H variant using antisense asymmetric oligonucleotide templates resulted in the correct knock-in of both variant and PAM-site change in approximately 2% of sequencing reads.

The variant of interest in *ripor2* is a 12-nucleotide deletion, which in theory can directly result from RNP activity followed by non-homologous end-joining (NHEJ) DNA repair, as CRISPR-Cas9 is known for the ability to introduce small insertions and deletions. However, Sanger sequencing traces of the majority of *ripor2* del12 RNP injected embryos indicate that this is not the case as the typical overlapping peaks start 3 nucleotides upstream of the intended 12-nt deletion (Figure 1D).

### 2.3. Knock-Down of the Non-Homologous End-Joining Pathway May Improve Knock-in Efficiency

We previously described the generation of a humanized zebrafish knock-in model for the pathogenic deep-intronic c.7595-2144A>G variant in human *USH2A*, resulting in the inclusion of a pseudo-exon (PE40). We applied a morpholino (PMO)-induced knock-down of Ku70, a critical component of the NHEJ pathway encoded by the *xrcc6* gene [13], to presumably improve knock-in efficiency [15]. The knock-in of the pseudo-exon was only observed upon co-injection of CRISPR-Cas9 components, template and the *xrcc6*-targeting PMO, suggesting that the suppression of Ku70 expression had a positive effect on the knock-in efficiency. As this positive effect was not quantified at the time, we investigated the potential improvement in knock-in efficiency of the target variants in *ush2a*, *ripor2*, and *tp53* after Ku70 knockdown. The *xrcc6*-targeting PMO, delivered into fertilized zebrafish cell together with the sgRNA-Cas9 RNP complexes, was previously titrated at 1.5 pg per embryo [15]. For each replicate injection, one or multiple pools of 20–25 embryos were sampled for genomic analysis of knock-in events at 1 dpf, and pools of 2 dpf embryos were sampled to assess PMO-induced alternative splicing of *xrcc6*. Presence of alternatively-spliced *xrcc6* transcripts was confirmed for all replicate injections that were included in this study. An example of the alternative *xrcc6* splicing is shown Figure 3A. Injection of the PMO-induced skipping of *xrcc6* exon 2 or the (partial) retention of intron 2, as can be observed in all three PMO-injected conditions, but not in uninjected controls. The three major alternatively-spliced transcripts were sequence verified and all contain premature translation termination codons. We next amplified the knock-in target regions from the genomic DNA of the zebrafish embryos injected with RNP and HDR template, with or without *xrcc6*-targeting PMO, and quantified the frequency of knock-in reads using ion torrent sequencing. To allow the comparison of our results with previously published studies, we quantified the percentage of correct knock-in reads that also included the PAM variant.

To our surprise, addition of the *xrcc6*-targeting PMO to the injection mixture increased the percentage of knock-in reads only for the *ripor2* del12 variant (Figure 3B). Since we observed a high variation in knock-in efficiency between the different replicate injections, we also visualized the change in knock-in efficiency between injections with and without PMO performed in the same clutch of eggs (Figure 3C). Three out of the four replicate injections with the *ripor2* del12 RNPs and *xrcc6*-targeting PMO resulted in a nearly two-fold increase in knock-in efficiency. For the *ush2a* C771F variant, two out of four replicates showed a marked decrease in knock-in efficiency upon addition of the *xrcc6*-targeting PMO to the injection mixture. For the *tp53*, co-delivery of the *xrcc6*-targeting PMO led to nearly two-fold increase in knock-in efficiency in one replicate, but had little effect in the replicate injection. To compare the effect of the *xrcc6*-targeting PMO on knock-in efficiency in an unbiased manner, and limit the influence of experimenter biases, skill and timing of injections as much as possible, all conditions with and without addition of PMO were injected by the same experimenter in one-cell-stage embryos from the same clutch. Injections for each target were furthermore conducted by different, skilled researchers, and the order of injections (with or without *xrcc6*-targeting PMO) was randomized between replications. Despite this careful approach, the gain in knock-in efficiency after co-injection of the *xrcc6*-targeting PMO is variable and appears to differ between targets.

### 2.4. Correlation between Somatic Knock-In Efficiency Shortly after Injection, and Germline Transmission in Adult Zebrafish

To establish stable knock-in zebrafish lines, it is of key importance that the mutations identified in injected zebrafish embryos have also occurred in the germline of the zebrafish. To increase the chances of successfully identifying injected adult fish with the variant of interest in the germline, and limit the number of potential founder fish to be screened, a high somatic knock-in efficiency is advantageous. Injecting very early in the zebrafish zygote can, in theory, lead to a complete somatic knock-in embryo. In practice, this is virtually impossible.

We screened injected adults for the presence of the variant of interest in their germline by breeding them with wildtypes, and analyzing the DNA of their offspring, to shed light on the correlation between somatic knock-in efficiency and the percentage of founder fish. For the *ush2a* C771F variant, we screened 10 potential founders, and identified germline transmission of the in C771F variant in 3 of them (Table 1). We identified germline transmission of the 12 bp deletion in *ripor2* in 5 out of 11 injected adults that we screened. These adults were raised from injected embryos in which we identified the *ush2a* C771F variant and *ripor2* del12 variant in respectively 3.4% and 8.6% of reads (Table 1), indicating that somatic knock-in efficiencies in >3% of reads allow for swift identification of founders. Note that all values on germline transmission include both knock-in events with and without in *cis* knock-in of the PAM site variant, as our objective was to develop knock-in zebrafish lines for the patient-specific variants in *USH2A* and *RIPOR2*. Stable knock-in lines were established from *ush2a* C771F and *ripor2* del12 founder fish. Genomic lesions were not identified at the predicted off-target sites in the F1 fish from which we established the stable *ush2a* C771F line (Reurink et al., in prep.).

## 3. Discussion

We here present our efficient approach to develop knock-in zebrafish models to model human disorders, illustrated for two pathogenic variants that were identified in patients with hereditary sensory deficits. Our method combines previously described advantages of Cas9-gRNA ribonucleoprotein (RNP) complex delivery [21], and antisense asymmetric DNA oligo knock-in templates [4]. With this method, we observed correct knock-in of the variant of interest in 3.4–18.0% of sequencing reads in DNA extracted from injected embryos. We furthermore show that this approach is also highly efficient to knock-in a small in-frame deletion, which to the best of our knowledge was not shown before. Knockdown of Ku70 further increased knock-in efficiency for the 12-nt deletion. Raising embryos with somatic knock-in mutations in 3–8% of sequencing reads lead to germline transmission of the variant of interest for 30–45% of the injected larvae. The detailed technical and procedural insights of this study, summarized in Figure 4, provide a valuable framework for the efficient development of knock-in zebrafish models. Phenotypic analyses of the established genetic zebrafish models are ongoing.

For a long time, modeling of human inherited disorders in zebrafish relied on the use of loss-of-function mutations that were obtained from forward genetic screenings, and more recently from CRISPR-Cas9-mediated genome editing [22,23,24]. However, with the large variation in disease-associated genetic variants and genetic disease mechanisms, the ability to introduce patient- or disease-specific variants in the zebrafish genome becomes increasingly important. When modeling disease-associated variants, there is little to no flexibility in design of the CRISPR-Cas9 strategy. Assessing and optimizing knock-in efficiency based on introduced restriction sites or phenotypic changes, often used in CRISPR-Cas9 optimization studies, is not an option when modeling human diseases.

The possibility of introducing variants of interest strongly correlates with the ability to direct Cas9 to cleave the DNA in close proximity to the target nucleotide(s) [7]. The ability of cleavage close to the target nucleotide is largely determined by the requirement of Cas9 for a specific protospacer adjacent motif (PAM): NGG. Especially when relying on commercially available CRISPR-associated proteins for RNP injection, it is not yet possible to exploit the growing catalog of nucleases with different PAM site preferences [25]. By chance, the design of the guide RNAs to knock in the 12-nucleotide deletion in *ripor2* was easy, as the target region contains two NGG triplets within the deletion region, and NGG triplets immediately upstream and downstream of the intended deletion. The situation was more complicated for the introduction of the c.2312G>T (p.(C771F)) variant in *ush2a*, as few NGG triplets were available in close proximity of the target variant. The most suitable option was a sgRNA with predicted off-target effects. All three predicted off-target sites had multiple mismatches with the sgRNA, which we considered an acceptable tradeoff since the use of alternative sgRNAs was expected to significantly decrease knock-in efficiency. We indeed obtained an F1 fish with the c.2312G>T variant, and no detectible off-target effects, from which we were able to generate a stable knock-in line in which we can assess the molecular and functional consequences of the variant. The targeting dilemma of the *ush2a* c.2312G>T variant perfectly reflects the limitations that can be encountered when introducing disease-associated variants in the zebrafish genome, and remains a challenge for the routine generation of zebrafish knock-in models to assess the pathogenicity of newly identified genetic variants in the context of genetic diagnostics.

The efficient introduction of genetic variants by homology-directed repair (HDR) strongly depends on the efficiency of sgRNA-directed cleavage of target DNA by Cas9. While the available algorithms for sgRNA design are generally very accurate at predicting sgRNA efficiency, it is advisable to assess this prior to investigating potential knock-in events. While many studies employ T7 Endonuclease digestions or Next-Generation Sequencing (NGS) to determine RNP efficiency [4,21], we prefer to assess RNP efficiency by using High Resolution Melting (HRM) analysis [19]. This is a quick and effective method to visualize Cas9-induced mutations (insertions and deletions) via the typical heteroduplex signatures in the melting pattern of a PCR product. In our experience, efficient sgRNAs can introduce heteroduplex signatures in 70–100% of RNP-injected embryos. Although in-vitro transcribed sgRNAs were used for the RNP injections of the current study, we recently switched to commercially produced sgRNAs as they have become more cost-effective.

For both addressed variants, we were able to continue the development of the knock-in zebrafish with highly efficient RNPs according to these criteria. We opted to combine two previously published improvements in zebrafish CRISPR technology, the use of sgRNA-Cas9 RNP complexes, and antisense asymmetric knock-in template oligos [4,21]. After the injection of both *ush2a* and *ripor2* sgRNA-Cas9 RNP complexes and corresponding template oligos, we observed on average less than 20% of wildtype reads in the genomic DNA of pools of injected embryos, confirming the high genome-editing efficacy of RNP delivery in zebrafish embryos. We furthermore identified over 5% and 12% of correct knock-in reads of respectively the *ush2a* c.2312G>T and *ripor2* del12 variant. These values are relatively high compared to other studies employing short, single stranded oligonucleotide templates (e.g., 1.92% in [4] and 2–4% in [4,26]). Especially when considering the fact that we introduced disease-specific variants, and encountered some limitations in sgRNA design for the introduction of the p.Cys771Phe variant in zebrafish usherin. To provide more perspective, we also injected the *tp53* sgRNA-Cas9 RNPs and R217H knock-in template that was previously described by Prykhozhij et al., and observed lower percentages of wildtype reads, and higher percentages of correct knock-in reads, as compared to their publication in which *Cas9* mRNA was used [4].

During the course of our studies, Bai and colleagues published that long single strand oligonucleotide templates can provide similar values of knock-in efficiency as we obtained here [27]. It is important to note that these long single strand oligonucleotides are indeed longer than our antisense asymmetric oligos, but still far shorter than the (circular) double-stranded templates previously employed by us and others [3,15,28]. In addition, Bai et al. first identified a highly efficient sgRNA-Cas9 combination to develop disrupt a pigmentation-associated gene, which they later repaired by HDR, and therefore this study may not reflect the sequence-dependent challenges encountered with introducing disease-associated variants.

One of the few studies that do describe the generation of knock-in zebrafish models to study the effect of disease-associated genetic variants makes use of symmetric oligonucleotide templates. While no data is provided on somatic knock-in efficiency, this approach does appear effective as germline transmission is found in 4–21% of injected zebrafish [5]. While many publications address the improvement of somatic knock-in efficiency in injected embryos, the chances of identifying founder fish that pass the variant of interest to their progeny is the most important factor for the swift development of knock-in zebrafish models. In this respect, the aforementioned study already offers useful germline transmission values. With our approach, we identified germline transmission of the variant of interest in 30% (*ush2a*) and 45% (*ripor2*) of injected adults, whereas somatic knock-in variants were identified in respectively 3.4% and 8.6% of sequences reads obtained from a subset of embryos from these specific injection rounds. With these particularly high germline transmission values obtained in this study, only few potential founder fish have to be screened, and raising high numbers of injected embryos appears unnecessary. Both the efficiency and timing of genome-editing are likely to be major factors in the chances of introducing the variant of interest in the germline. Although more studies are needed to strengthen the correlation between somatic knock-in efficiency and germline transmission of the variant of interest, our study provides some initial insight in this relationship.

The HDR and non-homologous end-joining (NHEJ) pathway are competitive DNA repair processes, and different approaches to shift the balance towards HDR have been tested in various species [29]. It was previously shown that the zebrafish Ku70 protein plays an important role in the repair of double strand DNA breaks [13]. We previously suggested that lowering the NHEJ pathway via PMO-induced knockdown of Ku70 expression could improve knock-in efficiency of the variants of interest [15]. Although addition of the *xrcc6*-targeting PMO to the *ripor2* del12 RNP/template mixture induced a two-fold increase correct knock-in events in three out of four batches of injected embryos, the effect of the *xrcc6*-targeting PMO was not consistent for all targets. We observed in increased knock-in efficiency for only two out of six replicate injections of the *ush2a* C771F and *tp53* R217H targets combined. Possibly, PMO-induced repression of Ku70 expression may be more effective in the generation of small deletions such as the *ripor2* del12 variant than for nucleotide substitutions. Due to the very early moment in zebrafish development at which we aim to lower the NHEJ pathway, it was not possible the determine if Ku70 protein levels and the activity of the NHEJ pathway were indeed decreased. Additionally, our splice-modulation PMO may leave some maternally contributed *xrcc6* transcripts intact. Hence, a translation-blocking PMO could more clearly improve HDR efficiency, but this its functionality more difficult to assess after microinjection than a splice-modulation PMO. While aberrant splicing of the Ku70-encoding *xrcc6* gene was confirmed for embryos sampled from all injections included in the analysis, the level of aberrant splicing was variable. It is also interesting to speculate on the effect of Ku70 knockdown on off-target genome editing. In theory, increasing the HDR pathway will also increase the chances that an off-target lesion is repaired based on the second allele, rather than via the NHEJ pathway. However, investigating the effect of Ku70 knockdown on off-target lesions is premature. We currently do not know how much knockdown of Ku70 expression is needed to drive the repair of Cas9-induced DNA lesions towards HDR, and whether the variable effect of the *xrcc6*-targeting PMO is mainly a dosage issue. In case of low somatic knock-in events—e.g., in <2% of sequencing reads obtained from a pool of injected embryos—it is worthwhile to investigate if co-delivery of the *xrcc6*-targeting PMO with the CRISPR-mixture can increase knock-in efficiency for that particular target. Especially since we show the chances of identifying founder fish are very favorable with a somatic knock-in efficiency of >3% of sequencing reads.

Despite our best effort to rule out all bias in this study, timing of the injection of the first cell of the zebrafish embryo may play a significant role in the knock-in efficiency. The earlier the sgRNA-Cas9 RNP complexes and template are present in the first cell of the zebrafish embryo, the higher the chances of knock-in events before the first DNA replication. While we aim to start RNP injections in the zygotic zebrafish cell within 15 min after parental zebrafish are placed together to produce eggs, the actual starting moment of injections remains difficult to standardize.

We aim to establish knock-in zebrafish models for the patient-specific target variants, and the PAM-site variants are just a means to reduce the chances of Cas9 re-cleaving a knock-in allele [5]. For both *ush2a* C771F and *tp53* R217H RNP injected larvae, the majority of correct reads did contain the PAM site variant. The knock-in alleles without PAM site variant likely result from crossover during homology directed repair [14]. This notion is confirmed by the fact that we also observed sequencing reads with the PAM site variant, but not the variant of interest. Sequencing of genomic DNA from *ripor2* del12 RNP injected larvae did not provide any indication that the 12-nucleotide deletion resulted from the injection of the RNPs alone. Despite the silent nature of the introduced PAM site variants, we prefer knock-in zebrafish without the additional PAM-site variant as disease models. Although unlikely, the PAM site variant may introduce a rare codon that may alter translational efficiency of the target gene or pre-mRNA [30]. Especially for the development of zebrafish models with gain-of-function alleles, or studies aimed to elucidate the pathogenicity of a variant of unknown significant (VUS) identified in genetic diagnostics, this risk is best avoided it is remains virtually impossible to determine if these variants have any (cis) modifying effects on the gene of interest.

In conclusion, our use of sgRNA-Cas9 RNP complex injections, combined with antisense asymmetric template oligos, allows for a highly efficient knock-in of genetic variants associated with human diseases. We show that this approach is successful for both point mutations and small in-frame deletions. Likely, our approach will also be successful to introduce shorter and slightly larger genomic deletions. We recently also published as strategy using 2 sgRNA-Cas9 RNP complexes to introduce larger deletions in the zebrafish genome, such as entire exons [31]. Our approach for the introduction of nucleotide substitutions and small, in-frame deletions is summarized in Figure 4, where we provide our recommended criteria to assess RNP efficiency and knock-in efficiency to maximize the chances of identifying founder fish with minimal screening efforts. To easily replicate our strategy in other labs, we have included a detailed technical procedure in the Appendix A.

The majority of novel diseases-associated variants that are nowadays identified, are found in isolated cases or families. As such, modeling the functional effects of these variants in animal models is of key importance to provide these patients with a genetic diagnosis. We envision that our step-by-step description will enable the more frequent use of zebrafish to assess the pathogenicity genetic variants associated with inherited disorders, and the molecular mechanisms underlying these disorders. The *ush2a* C771F and *ripor2* del12 zebrafish lines will indeed be used to determine respectively the pathogenicity and the pathogenic mechanisms of these variants in ongoing studies.

## 4. Materials and Methods

### 4.1. Zebrafish Husbandry

Wild-type Tüpfel long fin zebrafish were bred and raised under standard conditions [32]. Both adult and larval zebrafish were kept at a light–dark regime of 14 h light: 10 h darkness. Adult zebrafish were fed twice daily with Gemma Micro (Skretting, Tooele, UT, USA) and live brine shrimp. All experiments were carried out in accordance with European guidelines on animal experiments (2010/63/EU). Zebrafish eggs were obtained from natural spawning and reared at 28.5 °C in E3 embryo medium (5 mM NaCl, 0.17 mM KCl, 0.33 mM CaCl_2_, and 0.33 mM MgSO_4_), supplemented with 0.1% methylene blue.

### 4.2. sgRNA Synthesis

Design of the sgRNAs was done using CRISPRscan (https://www.crisprscan.org (accessed on 29 November 2018)) and CHOPCHOP (https://chopchop.cbu.uib.no (accessed on 29 November 2018)) webtools with settings for Cas9 and the in-vitro T7 promotor. The sgRNAs were ordered as single stranded DNA oligonucleotides as previously described by Gagnon et al. [24], with the addition of 5′-CCGCTAGC-3′ to the target specific oligonucleotide to clamp the 5′end. Sequences are provided in Appendix A. Target-specific and constant oligonucleotides were annealed and complemented using Phusion High-Fidelity DNA Polymerase (#M0530S, New England BioLabs, Ipswich, MA, USA). The reaction mixture, containing 5 µM of each oligonucleotide, 0.2 mM dNTPs, 1× Phusion HF buffer and 10 units/mL Phusion polymerase, was incubated sequentially for 2 min at 98 °C, 10 min at 50 °C, and 10 min at 72 °C. Annealed sgRNA templates were purified using the GenElute PCR clean-up kit (#NA1020, Sigma-Aldrich, Amsterdam, The Netherlands) according to manufacturer’s instructions. Synthesis of sgRNAs was done using the MEGAshortscript™ T7 Transcription Kit (#AM1354, Invitrogen, Waltham, MA, USA) according to manufacturer’s instruction, using 200–400 ng of sgRNA template and overnight incubation at 37 °C. Finally, sgRNAs were purified using the MEGAclear™ Transcription Clean-Up Kit (#AM1908, Invitrogen, Waltham, MA, USA) according to manufacturer’s instructions, aliquoted and stored at −80 °C until use.

### 4.3. HDR Template Design

The asymmetric antisense oligonucleotide templates are comprised of a 90-nucleotide arm and a 36-nucleotide arm calculated from the Cas9 cleavage position. The short arm is complementary to the sequence upstream of the Cas9 cleavage position, the long arm complementary to the downstream region. Note that the sgRNA-Cas9 RNP complex can target either DNA strand, and cleavage position is always determined as 3–4 nucleotides upstream of the PAM site (NGG for S.p.Cas9). Variants of interest were introduced in the short arm, PAM site variants in the long arm. The 12-nt deletion *ripor2* template was designed by removing these specific 12 nucleotide from the template, effectively reducing the length of the short arm.

### 4.4. Zebrafish Embryo Microinjections

Injection mixtures containing 800 ng/µL Alt-R^®^ S.p.Cas9 Nuclease V3 (Integrated DNA Technologies, Coralville, IA, USA) sgRNA, 300 mM KCl and 20% (*v*/*v*) of phenol red solution (#P0290, Sigma Aldrich, Amsterdam, The Netherlands) were incubated for 5 min at 37 °C to allow sgRNA-Cas9 ribonucleoprotein complex formation. Final concentrations of sgRNA in the injection mixture were 81 ng/µL (*ripor2*), 90 ng/µL (*ush2a*), and 150 ng/µL (*tp53*). Subsequently, 1 nL of mixture was injected into zebrafish zygotes as previously described [33]. Single-stranded antisense asymmetric DNA oligonucleotide templates (Appendix A) containing the variants of interest were ordered from Integrated DNA technologies, and added to the injection mixtures at a final concentration of 1 µM. *xrcc6* splice-modulation morpholino oligonucleotide was purchased from Gene-Tools (Philomath, OR, USA) and added to the mixture at a final concentration of 1.5 ng/µL as previously described [15]. All injections were performed for >80 zygotes per condition, and only embryos with >80% survival in injected and uninjected conditions were included for subsequent analyses.

### 4.5. High-Resolution Melting Analysis

Injected and uninjected zebrafish embryos were collected individually at 1 dpf in 25 uL hotshot lysis buffer (25 mM NaOH, 0.2 mM EDTA), incubated at 95 °C for 20 min, and neutralized with 2.5µL 1 M Tris pH 8. Obtained embryo lysates were diluted 5–10× and used as input for high-resolution melting (HRM) analysis. HRM PCR was performed in a Quantstudio 3 Real-Time PCR System (Applied Biosystems, Waltham, MA, USA), using Phusion High-Fidelity DNA Polymerase (#M0530S, New England Biolabs, Ipswich, MA, USA) and 1× EvaGreen Dye (#31000, Biotium, Fremont, CA, USA). After PCR amplification, the HRM procedure was initiated by incubating the reaction at 95 °C for 1 min, followed by rapid cooling (4 °C/s) to 10 °C and a standard melt curve procedure (0.1 °C/s) during which data was collected. Primer sequences are provided in Appendix A.

### 4.6. Ion Torrent Sequencing and Analysis

DNA was extracted from pools of 20–25 sgRNA-Cas9 RNP complex-injected 1 dpf zebrafish embryos, or offspring from potential founder fish, using the QIAamp DNA Mini Kit (#51304, QIAGEN, Hilden, Germany). PCR products for Ion Torrent sequencing were prepared using Q5 High-Fidelity polymerase (#M0491L, New England BioLabs, Ipswich, MA, USA) and target-specific primers with or without barcodes for multiplex sequencing (Appendix A). PCR products from individual biological samples were amplified using different indexed primers and then pooled in equimolar amounts into sequencing samples. Barcoding was not used in the initial experiments where unique amplicons were pooled. The obtained sequencing reads were demultiplexed, mapped against wildtype specific Fasta files, and analyzed for knock-in events using the SEQNEXT software (JSI Medical Systems, Ettenheim, Germany). The percentage of knock-in reads was calculated by filtering reads to contain the knock-in events and correct flanking sequence for 20 bp up- and downstream of the variants of interest. The obtained sequencing reads contained an occasional mismatch with the reference sequence, but in light of the very low frequency of these variants (>0.01 of reads), known error rate of the Ion Torrent System, and distance to the Cas9 cleavage site, additional filtering was not applied.

## Figures and Tables

**Figure 1 ijms-22-09429-f001:**
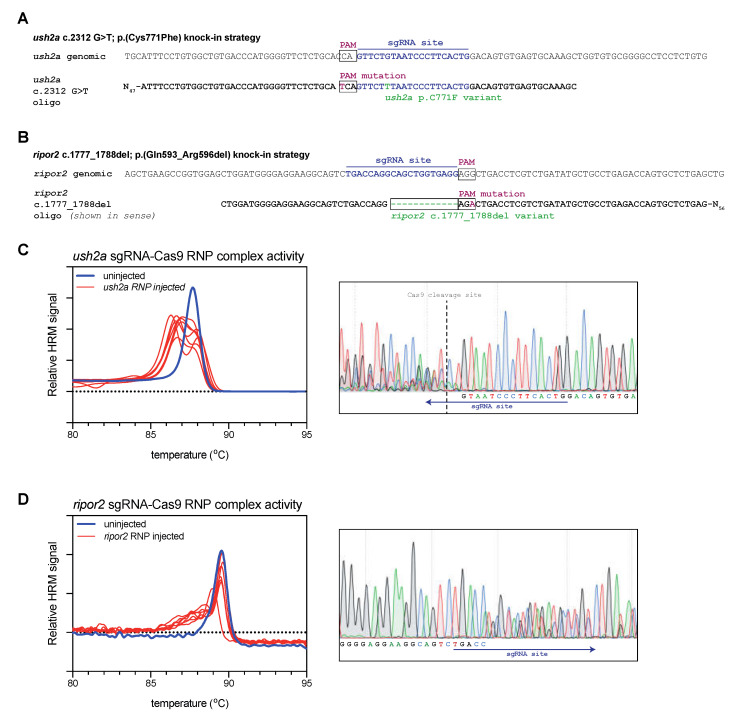
Study design and validation of sgRNA-Cas9 ribonucleoprotein complex efficiency. (**A**) Graphic representation of the target region in *ush2a* exon 13. The sgRNA (targeting the complementary strand) and protospacer-adjacent motif (PAM) are indicated and allow for Cas9 cleavage in close proximity of position c.2312. The asymmetric antisense template for homology directed repair contains the c.2312G>T variant and a silent PAM-site change. (**B**) Graphic representation of the target region in *ripor2* exon 14. The sgRNA is designed to direct Cas9 to cleave within the nucleotide sequence that we aimed to delete. The asymmetric antisense template for homology directed repair, containing the 12-nucleotide deletion and a silent PAM-site change, is shown in sense orientation. (**C**) High resolution melting (HRM) analysis profiles of individual embryos injected with the *ush2a* sgRNA-Cas9 ribonucleoprotein (RNP) complex. A representative example of the Sanger sequencing traces from a single embryo is shown on the right. The typical overlapping peaks occur around the indicated Cas9 cleavage site. (**D**) HRM analysis profiles of individual *ripor2* sgRNA-Cas9 ribonucleoprotein (RNP) complex injected embryos. The typical overlapping peaks in the sanger sequencing traces, shown on the right, are indicative of sgRNA-Cas9 RNP activity.

**Figure 2 ijms-22-09429-f002:**
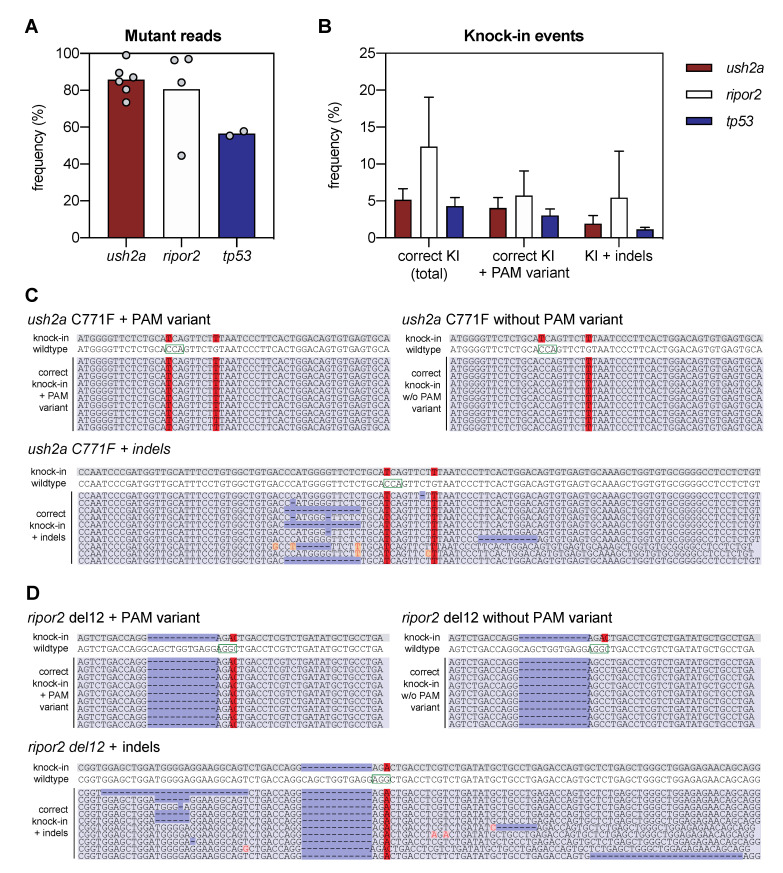
Next-generation sequencing analysis of knock-in efficiency after sgRNA-Cas9 RNP and HDR template delivery. (**A**) Percentage of mutant reads in the genomic DNA of pooled embryos after injection of sgRNA, Cas9, and asymmetric antisense oligonucleotide templates. Mutant reads are quantified as the percentage of all non-wildtype reads. Data is expressed as mean (bar) and values of individual replicates (scatterplot). (**B**) Knock-in efficiency after injection of sgRNA, Cas9 and asymmetric antisense oligonucleotide template. Knock-in (KI) reads are quantified as percentage of total reads with correct knock-in with and without the silent variant in the protospacer adjacent motif (PAM). The percentage of correct knock-in reads with additional variants is provided as KI + indels. Examples of correct knock-in reads with and without PAM variants and indels are provided for (**C**) *ush2a* C771F RNP injected embryos, and (**D**) *ripor2* del12 RNP injected embryos. The correctly-introduced nucleotide substitutions are indicated in red. Unintended nucleotide substitutions are indicated with a red font on orange background. Deletions are indicated in dark blue. The PAM sites are indicated on the wildtype sequences by a green box.

**Figure 3 ijms-22-09429-f003:**
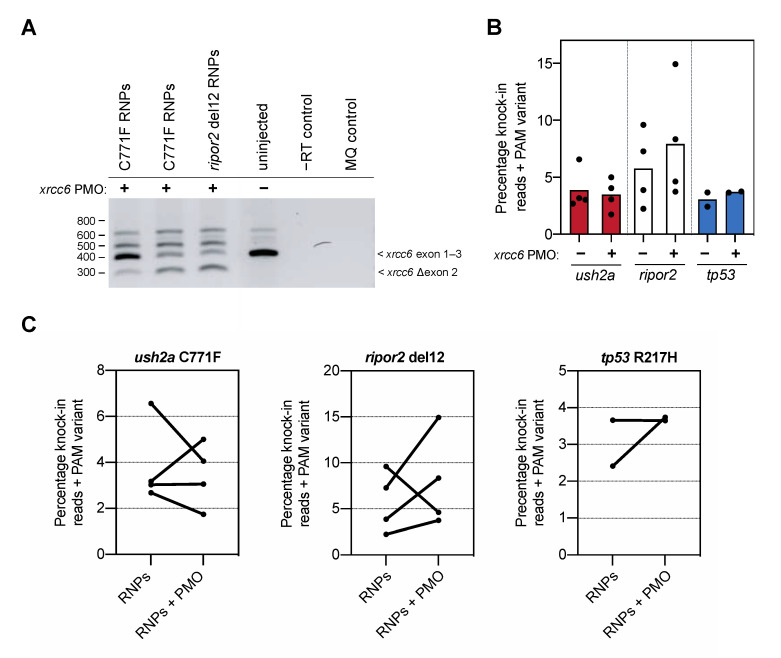
Knock-in efficiency after co-delivery of the *xrcc6*-targeting PMO. (**A**) Addition of the *xrcc6*-targeting PMO to the RNP injection mixtures results in alterative *xrcc6* splicing. Both intron retention and exon 2-skipping are observed upon RT-PCR analysis of the *xrcc6* transcript in PMO-injected embryos, but not control injected embryos. Replicate injections of C711F RNPs provide insight in variability of PMO-induced alterative *xrcc6* splicing. (**B**) Quantification of knock-in efficiency after sgRNA-Cas9 RNP and template injection with and without co-injection of the *xrcc6*-targeting PMO. For *ripor2*, possibly *tp53,* but not *ush2a*, an increase in knock-in events is observed after co-injection of sgRNA-Cas9 RNPs, template oligonucleotide and the *xrcc6*-targeting PMO. Results are expressed as mean (bars) and individual replicates (scatterplot). Only reads that include the PAM variant are included in the quantification to ensure that all quantified correct knock-in reads are the project of homology directed repair. (**C**) Comparison of changes in knock-in efficiency between embryos of the same clutch of eggs. For *ush2a* C771F RNPs, addition of the PMO to the injection mixture leads to variable outcomes in knock-in efficiency between the replicate injections. Three out of the four replicate injections of *ripor2* del12 RNPs with *xrcc6*-tarageting PMO display a higher knock-in efficiency, while for *tp53* the addition of the PMO leads to an increase in knock-in efficiency in one of the two replicate injections.

**Figure 4 ijms-22-09429-f004:**
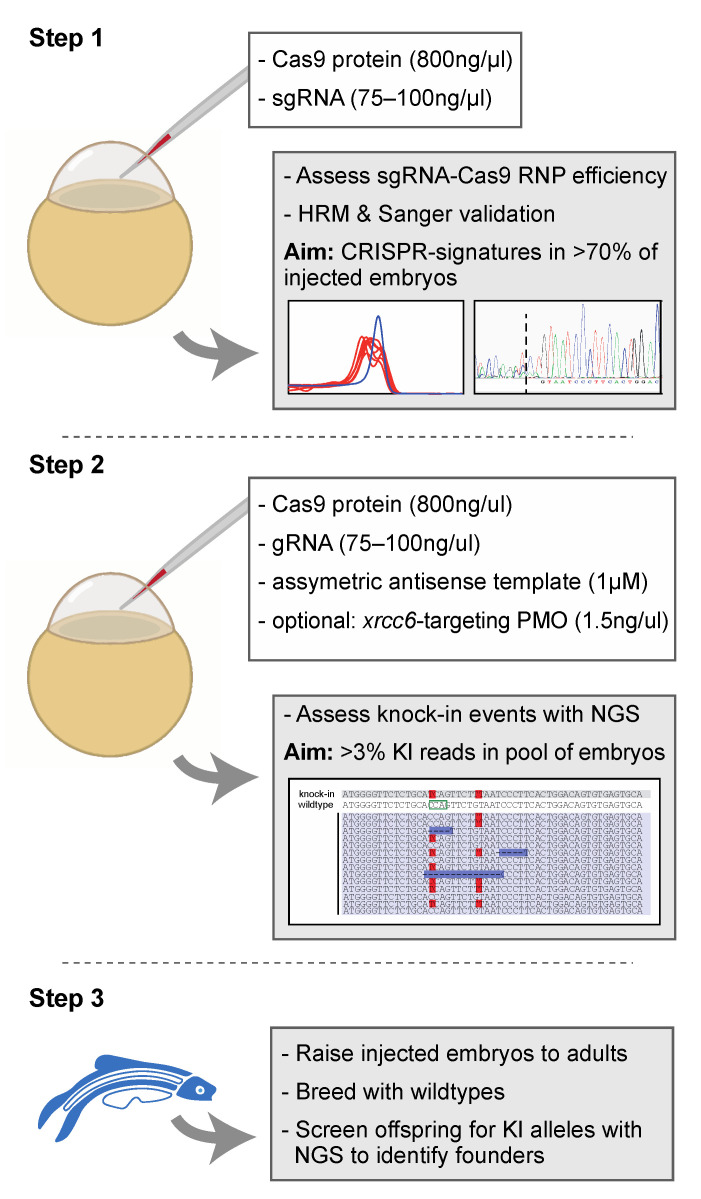
Proposed strategy for the efficient generation of knock-in zebrafish. Step 1 entails the verification of sgRNA-Cas9 efficiency at creating genomic lesions using high resolution melting (HRM) analysis. Step 2 concerns the injection of sgRNA-Cas9 ribonucleoprotein complexes and the asymmetric antisense oligonucleotide template with the variant of interest. Next generation sequencing (NGS) is recommended to determine knock-in efficiency. In the situation of low knock-in efficiency, knockdown of non-homologous end-joining protein Ku70 (by co-delivery of an morpholino antisense oligonucleotide (PMO), targeting *xrcc6*) may offer a means to improve knock-in efficiency. Finally, injected embryos with sufficient knock-in reads can be raised to adults and screened for founders with the variant of interest in the germline (step 3). The white boxes in step 1 and step 2 described the key components injected into the zebrafish zygote. A detailed protocol can be found in Appendix A.

**Table 1 ijms-22-09429-t001:** Somatic knock-in efficiency and subsequent germline transmission.

Target	KI Reads in Injected Embryos	Adult Fish Screened	Total Founders with Variant	Founders with PAM Variant
*ush2a* C771F	3.4%	10	3 (30%)	1 (10%)
*ripor2* del12	8.6%	11	5 (45%)	2 (18%)

## Data Availability

Not applicable.

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
