# Peer review of "Efficient Generation of Knock-In Zebrafish Models for Inherited Disorders Using CRISPR-Cas9 Ribonucleoprotein Complexes"

_ijms, 2021, doi:10.3390/ijms22179429_

Round 1

Reviewer 1 Report

The work provided by Vrieze et. al, describes an optimized approach for generating knock-in zebrafish models. While the manuscript specifically mentions the approach is for inherited disorders, it is applicable towards the generation of many desired alleles/mutations by the experimenter. Although it would have been nice to include images of zebrafish phenotypes (if any) that are due to the mutations, that is beyond the scope of this methods paper. As a methods paper, two modifications (for the purposes of those trying to replicate the procedure) are recommended: 1) a table summarizing the source of reagents used and 2) a detailed protocol to accompany Figure 4. The protocol should be a detail step by step guide to help those attempting to follow the work.

Reviewer 2 Report

In this manuscript, de Vrieze et al. reported a knock-in zebrafish models for study inherited disorders using CRISPR-Cas9 and proposed a strategy for efficient generation of knock-in zebrafish.

The manuscript is in a good shape, I only have some minor suggestions/questions.

  1. Page 5 line 185-186, the efficacy was actually estimated based on the distribution of Ion Torrent sequenced reads. Since the introduced deletion is only 12 bp length, there should be little or no amplification biases. The statement needs to reflect that.
  2. Page 5 line 181. The at least 50 reads per target region is per embryo or pooled samples?
  3. Fig 2 CD, only 10 reads were shown in the figure per region, where are the other reads?
  4. Only the targeted regions were sequenced, the authors need to discuss whether the introduction of PMO will increase non-target editing.
  5. The author only tried a 12-bp length deletion, in order to claim a new strategy for efficient generation of knock-in zebrafish, at least the authors need to discuss whether longer deletions and/or insertions will be benefited from this new strategy, in another word, the potential limitations.

Author Response

please see the attachment. Replies to both reviewers are combined in a single file (uploaded twice)
